# Human dynein–dynactin is a fast processive motor in living cells

Vikash Verma[1], Patricia Wadsworth[1,2]*, Thomas J Maresca[1,2]*

[1]Biology Department, University of Massachusetts, Amherst, United States; [2]Molecular and Cellular Biology Graduate Program, University of Massachusetts, Amherst, United States

## eLife Assessment

In this **valuable** technical report, Verma et al. provide **convincing** evidence that endogenously tagged dynein and dynactin form processive motor complexes that move along microtubules in living cells. Using quantitative fluorescence microscopy, they directly compare the stoichiometry and motility of these complexes to kinesin-1, revealing distinct transport behaviors and regulatory properties. This study offers key methodological and conceptual advance for understanding the dynamics of native motor proteins within the cellular environment and will be of interest to the cell biology community.

**Abstract** Minus-end directed transport along microtubules in eukaryotes is primarily mediated by cytoplasmic dynein and its cofactor dynactin. Significant advances have been made in recent years characterizing human dynein–dynactin structure and function using in vitro assays; however, there is limited knowledge about the motile properties and functional organization of dynein–dynactin in living human cells. Total internal reflection fluorescence microscopy of CRISPR-engineered human cells is employed here to visualize fluorescently tagged dynein heavy chain (DHC) and p50 with high spatio-temporal resolution. We find that p50 and DHC exhibit indistinguishable motility properties in their velocities, run lengths, and run times. The dynein–dynactin complexes are fast (~1.2 µm/s) and run for several microns (~2.7 µm). Quantification of the fluorescence intensities of motile puncta reveals that dynein–dynactin runs are mediated by at least one DHC dimer while the velocity is consistent with that measured for double dynein (two DHC dimers) complexes in vitro.

## Introduction

The eukaryotic microtubule (MT) cytoskeleton plays a critical role in the organization, positioning, and motility of organelles, mRNA, and proteins. Intracellular motility is mediated by molecular motor proteins that move cargoes along polarized MT tracks (for reviews see *Cason and Holzbaur, 2022*; *Reck-Peterson et al., 2018*). The kinesin superfamily includes motors that are responsible for plus-end directed transport, and others that contribute to minus-end motility and regulation of MT dynamics (*Hirokawa et al., 2009*). In contrast, cytoplasmic dynein 1 motor complexes (hereafter dynein) (*Canty et al., 2021*; *Pfister et al., 2005*) are exclusively minus-end directed. The dynein complex is composed of two catalytic heavy chains and additional light, light intermediate, and intermediate chains (*Carter et al., 2016*; *Cason and Holzbaur, 2022*). The isolated dynein motor complex displays predominantly diffuse motility along MTs in vitro, an observation that led to the identification of the dynactin complex, which is important for dynein motility (*Gill et al., 1991*; *McKenney et al., 2014*; *Schlager et al., 2014*; *Schroer and Sheetz, 1991*). In addition, a number of adaptor complexes that link dynein and dynactin to specific cargoes and activate the motor have been identified and

***For correspondence:**
patw@bio.umass.edu (PW);
tmaresca@umass.edu (TJM)

**Competing interest:** The authors declare that no competing interests exist.

characterized (*Canty et al., 2021*; *Cason and Holzbaur, 2022*; *Reck-Peterson et al., 2018*; *Splinter et al., 2012*).

Vesicular transport has been directly visualized by live-cell imaging in diverse cells including highly polarized axons (*Canty et al., 2021*). Motility of vesicles is bidirectional, both plus- and minus-end directed motors co-purify with cargoes, and adaptor complexes have been shown to associate with both kinesin and dynein motors (*Ali et al., 2023*; *Hancock, 2014*; *Hendricks et al., 2010*; *Maeder et al., 2014*; *Vale, 1987*). Together, these observations indicate that the number and activity of cargo-associated motors must be regulated to ensure efficient delivery to appropriate cellular destinations (*Cason and Holzbaur, 2022*). In vitro studies in which the number of motors bound to artificial cargoes can be precisely controlled demonstrate that cargoes with both plus and minus motors often stall or pause, or show directed motion that is distinct from either individual motor acting alone; however, once directed motion is initiated, reversals are infrequent (*Belyy et al., 2016*; *Derr et al., 2012*). Cryo-EM of dynein–dynactin complexes revealed that the stoichiometry of dynein relative to dynactin is determined by the activating adaptors that link them. Specifically, the adaptor BICD2 tends to favor single dynein, and the BICDR1 and HOOK3 tend to favor two dyneins assembled into a 'double dynein' complex (*Chaaban and Carter, 2022*; *Grotjahn et al., 2018*; *Urnavicius et al., 2018*; *Urnavicius et al., 2015*). Interestingly, the processivity and velocity of dynein–dynactin differ depending on the number of dyneins such that the double dynein complexes assembled with BICDR1 and HOOK3 had a higher frequency of processive motility events and moved faster than the single dynein complexes assembled with BICD2 (*Urnavicius et al., 2018*).

In contrast to the relative ease of visualizing motor proteins in vitro, the crowded three-dimensional intracellular environment poses a challenge for imaging and quantifying motility events in living cells. Puncta exhibiting bidirectional movement on MTs and comet-like motile events that co-localized with EB1 were observed in human cells over-expressing a GFP-tagged dynein intermediate chain, supporting a role for dynein in MT plus-end tip-tracking and vesicle motility (*Kobayashi and Murayama, 2009*). Quantification of organelle motility in the highly polarized filamentous fungus *Ustilago maydis* expressing endogenously tagged fluorescent dynein revealed that binding of a single dynein to an anterograde-directed early endosome was sufficient to trigger directional reversal of its transport (*Schuster et al., 2011*).

In this study, HeLa cells expressing CRISPR/Cas9-modified dynein heavy chain (DHC) or the p50 subunit of dynactin were visualized using high-resolution fluorescence microscopy approaches to gain insights into the behaviors of dynein and dynactin in living cells. In doing so, we were able to directly observe and quantify core motility parameters of dynein–dynactin while analyses of their fluorescence intensities relative to a known standard were informative of the stoichiometry of dynein in motile dynein–dynactin complexes in proliferating human cells.

## Results and discussion

Human HeLa cells were engineered using CRISPR/Cas9 to insert a cassette encoding FKBP and EGFP tags in frame at the 3′ end of the dynein heavy chain (DYNC1H1) gene (*Figure 1—figure supplement 1A*). A clonal DHC-EGFP-expressing HeLa cell line was generated and subjected to live-cell fluorescence imaging to directly visualize DHC activities in interphase cells (*Figure 1—figure supplement 1B*). The most striking dynein behavior was tip-tracking on polymerizing MTs as an abundance of DHC comets traveling at the speed of MT polymerization (*Cassimeris et al., 1988*; *Rusan et al., 2001*; *Walker et al., 1988*) were readily observed by both spinning disc confocal microscopy and total internal reflection fluorescence microscopy (TIRFM) (*Figure 1A, B*, *Figure 1—video 1*, *Figure 1—video 2*). SiR-Tubulin was next introduced to visualize DHC localization and behaviors relative to MTs (*Figure 1C*). Interestingly, because SiR-Tubulin is a docetaxel derivative, its addition suppressed plus-end MT polymerization resulting in a significant reduction in the DHC tip-tracking population and a much clearer view of a different population of MT-associated DHC puncta (*Figure 1C*). The SiR-Tubulin-treated cells were subjected to two-color TIRFM, and DHC-EGFP puncta were clearly observed streaming on SiR-Tubulin-labeled MTs, which was especially evident on MTs that were pinned between the nucleus and the plasma membrane (*Figure 1—video 3*). DHC puncta were next visualized with higher temporal resolution by acquiring single color TIRFM time-lapses of the EGFP channel at a rate of 5 frames per second (fps) (*Figure 1D*, *Figure 1—video 4*). The TIRFM time-lapses were assessed by eye and several parameters of the motile puncta were measured using kymographs

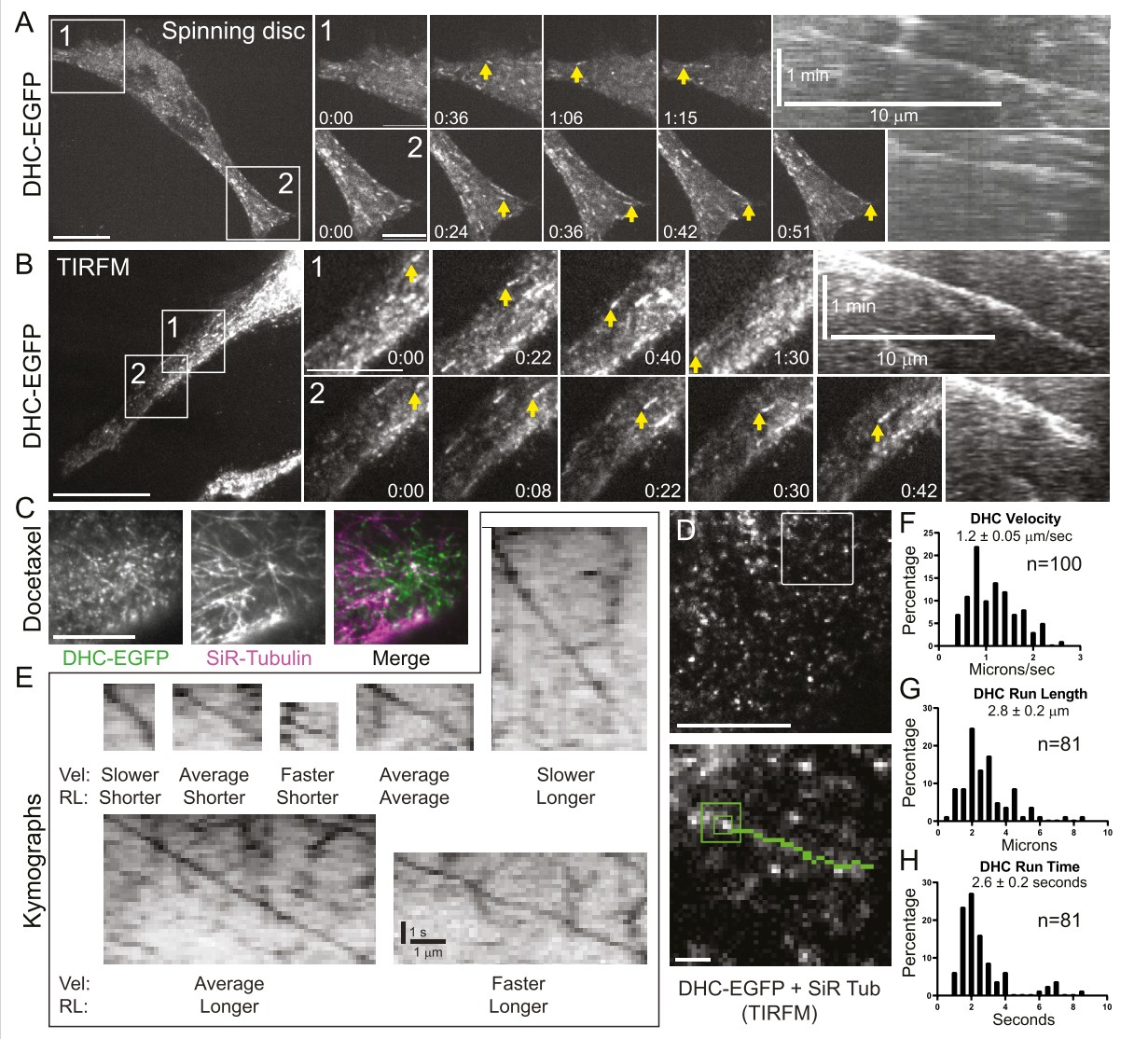

**Figure 1.** Dynein tip-tracks on polymerizing microtubules (MTs) and walks processively at high velocities in interphase HeLa cells. (**A**) Representative spinning disc confocal time-lapse of a DHC-EGFP-expressing HeLa cell showing robust tip-tracking. Zoomed views of the numbered boxed regions are shown to the right with kymographs of the tip-tracking events highlighted with yellow arrows in the zoomed panels. (**B**) Representative total internal reflection fluorescence microscopy (TIRFM) time-lapse of a DHC-EGFP-expressing HeLa cell showing the tip-tracking population. Zoomed views of the numbered boxed regions are shown with kymographs of the tip-tracking events highlighted with yellow arrows. (**C**) Still frame from a representative TIRFM time-lapse of a DHC-EGFP-expressing HeLa cell treated with SiR-Tubulin. In the merge image, DHC is green and Sir-Tubulin labeled MTs are magenta. (**D**) Still frame from a representative high temporal resolution (5 fps) TIRFM time-lapse of a DHC-EGFP-expressing HeLa cell treated with SiR-Tubulin. Boxed region is shown as a zoomed inset in the lower panel with the track of a motile DHC puncta highlighted in green. (**E**) Representative kymographs of motile puncta spanning the range of measured velocities (Vel) and run lengths (RL). (**F**) Distribution of DHC velocities (*n* = 100 puncta). (**G**) Distribution of DHC run lengths (*n* = 81 puncta). (**H**) Distribution of DHC run times (*n* = 81 puncta). Scale bars, 10 µm (**A–D**); 1 µm (all insets), 10 µm (horizontal); 1 min (vertical) in the kymographs in A and B; and 1 µm (horizontal); 1 s (vertical) in E. Displayed times are min:s.

The online version of this article includes the following video, source data, and figure supplement(s) for figure 1:

**Source data 1.** Excel spreadsheet containing the underlying data and numerical values for plots in *Figure 1*.

**Figure supplement 1.** Characterization of DHC-EGFP CRISPR knock-in cells.

**Figure 1—video 1.** Spinning disk confocal time-lapse of a DHC-EGFP CRISPR-engineered interphase HeLa cell.
https://elifesciences.org/articles/94963/figures#fig1video1

**Figure 1—video 2.** Total internal reflection fluorescence microscopy (TIRFM) time-lapse of a DHC-EGFP CRISPR-engineered interphase HeLa cell.
https://elifesciences.org/articles/94963/figures#fig1video2

**Figure 1—video 3.** Total internal reflection fluorescence microscopy (TIRFM) time-lapse of a DHC-EGFP (green) CRISPR-engineered interphase HeLa

*Figure 1 continued on next page*

*Figure 1 continued*

cell treated with SiR-Tubulin (magenta).

https://elifesciences.org/articles/94963/figures#fig1video3

**Figure 1—video 4.** High temporal-resolution (5 frames per second) total internal reflection fluorescence microscopy (TIRFM) time-lapse of a DHC-EGFP CRISPR-engineered interphase HeLa cell treated with SiR-Tubulin.

https://elifesciences.org/articles/94963/figures#fig1video4

(*Figure 1E*). The average DHC puncta moved at 1.2 ± 0.05 (mean ± SEM) μm/s over a distance of 2.8 ± 0.2 μm for 2.6 ± 0.2 s (*Figure 1F–H*). While some motile puncta appeared to switch 'tracks' at MT intersections, which was inferred from observations of sudden high angle turns, directional switches on the same MT were infrequent (~3% of runs).

Activation of dynein motility requires its interaction with the dynactin complex (*Chowdhury et al., 2015*; *Grotjahn et al., 2018*; *Urnavicius et al., 2018*; *Zhang et al., 2017*). CRISPR/Cas9-based genomic engineering was used to insert the cassette encoding FKBP and EGFP tags at the 3′ end of the p50/dynamitin gene and a clonal p50-EGFP-expressing HeLa cell line was subjected to live-cell fluorescence imaging to visualize dynactin in interphase cells (*Figure 2—figure supplement 1A–C*). Consistent with prior observations of dynactin localization (*Vaughan et al., 1999*; *Vaughan et al., 2002*) and like dynein, dynactin exhibited robust tip-tracking activity on growing MT plus-ends that was clearly visualized via both spinning disc confocal imaging and TIRFM (*Figure 2A, B*; *Figure 2—video 1*, *Figure 2—video 2*). Suppression of plus-end polymerization dynamics upon introduction of SiR-Tubulin caused a loss of the tip-tracking pool of dynactin, which made a second pool of motile, MT-associated puncta of p50 more pronounced especially when visualized with TIRFM (*Figure 2C*, *Figure 2—videos 3; 4*). The p50-EGFP-expressing cells were then subjected to single color TIRFM at 5 fps and kymograph analysis to measure the motility parameters of the dynactin complex (*Figure 2D*). The mean velocity of the p50 puncta was 1.2 ± 0.07 μm/s (*Figure 2E*) while the mean run length was 2.6 ± 0.2 μm (*Figure 2F*) and the mean run time was 2.2 ± 0.2 s (*Figure 2G*). Similar to dynein, dynactin was observed to switch MT 'tracks' but infrequently switched direction. In comparing the motility parameters of the motile dynein and dynactin puncta, their velocities (*Figure 3A*), run lengths (*Figure 3B*), and run times (*Figure 3C*) were statistically indistinguishable. Thus, we inferred that the motile population of dynein was associated with the dynactin complex.

The fluorescence intensities of motile DHC and p50 puncta were next compared to motile EGFP-tagged kinesin-1 dimers (*Figure 3—figure supplement 1A–C*) as a standard to assess the stoichiometries of the dynein–dynactin complexes. The intensities of motile DHC and p50 puncta were compared to kinesin-1-EGFP expressed in either *Drosophila melanogaster* S2 cells (*Figure 3—figure supplement 2A–C*) or HeLa cells (*Figure 3D–F*). Interestingly, the intensity of motile DHC-EGFP puncta was not statistically significantly different from the intensity of motile kinesin-1-EGFP dimers; however, the intensity of motile p50 puncta had a mean fluorescence intensity that was about half that of the motile kinesin-1-EGFP puncta (*Figure 3G*, *Figure 3—figure supplement 2D*). Thus, the motile dynein and dynactin complexes visualized here via high-speed TIRFM were comprised, on average, of two EGFP-tagged DHC molecules and a single p50-EGFP molecule. When considering how these values relate to physiological stoichiometries of the motile dynein–dynactin complexes, it is important to note that both the DHC-EGFP and p50-EGFP HeLa cell line clones are heterozygotes (*Figure 3H, I*). Dynein motility requires DHC dimerization, and if there is an equal likelihood of EGFP-tagged and untagged DHC incorporation into a functional dimer, then the motile dynein puncta we visualized would contain two DHC dimers. We were unable to assess the relative expression levels of tagged versus untagged DHC by western blot due to the very large size of DHC (>500 kDa) relative to the EGFP tag. However, we can conclude from our data that there is at least one DHC dimer in the motile puncta. Interestingly, western blotting of cell extract prepared from the p50-EGFP clone revealed that the cells expressed a ~5- to 6-fold molar excess of the untagged p50 compared to p50-EGFP (*Figure 3I*). Differential allele regulation has been observed for endogenously tagged proteins (*Mann and Wadsworth, 2018*; *Roberts et al., 2017*), suggesting that regulation of gene expression may help avoid deleterious effects when cells can only tolerate a fraction of the total protein pool being modified. The relative levels of EGFP-tagged versus untagged p50 most likely explain why motile p50 puncta only had a single EGFP molecule despite the fact that the dynactin complex is known to have four copies of p50

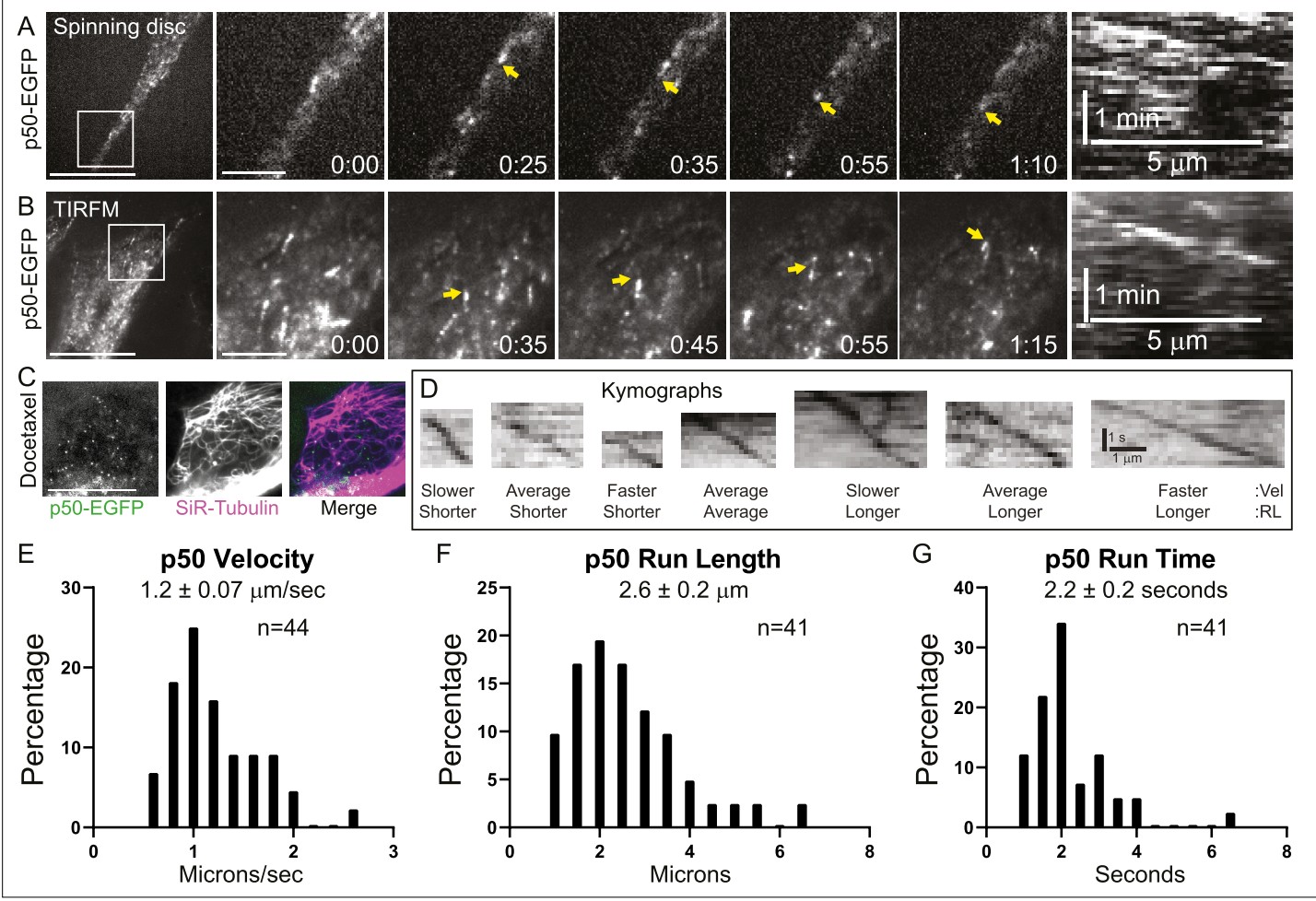

**Figure 2.** The dynactin complex component p50 tip-tracks on polymerizing microtubules (MTs) and walks processively at high velocities in interphase HeLa cells. (**A**) Still frames from a representative spinning disc confocal time-lapse of a p50-EGFP-expressing HeLa cell. A zoomed view of the boxed region is shown with a kymograph of the tip-tracking event highlighted with the yellow arrow. (**B**) Still frames from a representative total internal reflection fluorescence microscopy (TIRFM) time-lapse of a p50-EGFP-expressing HeLa cell showing the tip-tracking population. A zoomed view of the boxed region is shown with a kymograph of the tip-tracking event highlighted by the yellow arrow. (**C**) Still frame from a representative TIRFM time-lapse of a p50-EGFP-expressing HeLa cell treated with SiR-Tubulin. In the merge image, p50 is green and Sir-Tubulin labeled MTs are magenta. (**D**) Representative kymographs of motile p50 puncta spanning the range of measured velocities (Vel) and run lengths (RL). (**E**) Distribution of p50 velocities (*n* = 44 puncta). (**F**) Distribution of p50 run lengths (*n* = 41 puncta). (**G**) Distribution of p50 run times (*n* = 41 puncta). Scale bars, 10 μm (**A–C**); 1 μm (all insets), 5 μm (horizontal); 1 min (vertical) in the kymographs in A and B; and 1 μm (horizontal); 1 s (vertical) in D. Displayed times are min:s.

The online version of this article includes the following video, source data, and figure supplement(s) for figure 2:

**Source data 1.** Excel spreadsheet containing the underlying data and numerical values for plots in *Figure 2*.

**Figure supplement 1.** Comparison of the p50-EGFP and DHC-EGFP CRISPR knock-in cells.

**Figure supplement 1—source data 1.** Excel spreadsheet containing the underlying data and numerical values for plots in *Figure 2—figure supplement 1*.

**Figure 2—video 1.** Spinning disk confocal time-lapse of a p50-EGFP CRISPR-engineered interphase HeLa cell.
https://elifesciences.org/articles/94963/figures#fig2video1

**Figure 2—video 2.** Total internal reflection fluorescence microscopy (TIRFM) time-lapse of a p50-EGFP CRISPR-engineered interphase HeLa cell.
https://elifesciences.org/articles/94963/figures#fig2video2

**Figure 2—video 3.** Total internal reflection fluorescence microscopy (TIRFM) time-lapse of a p50-EGFP (green) CRISPR-engineered interphase HeLa cell treated with SiR-Tubulin (magenta).
https://elifesciences.org/articles/94963/figures#fig2video3

**Figure 2—video 4.** Total internal reflection fluorescence microscopy (TIRFM) time-lapse of a p50-EGFP (green) CRISPR-engineered interphase HeLa cell treated with SiR-Tubulin (magenta).
https://elifesciences.org/articles/94963/figures#fig2video4

(*Eckley et al., 1999*; *Urnavicius et al., 2015*). Thus, we conclude that the motile dynactin complexes visualized here are typically comprised of one p50-EGFP molecule and 3 untagged copies of p50.

While several groups have recently imaged endogenously tagged dynein in mitotic HeLa cells and iNeurons (*Fellows et al., 2024*; *Ide et al., 2023*), to our knowledge, this is the first direct visualization of motile, endogenously tagged human dynein–dynactin complexes in proliferating, non-neuronal cells. In human interphase cells, two distinct motile populations of dynein–dynactin are observed: one that tip-tracks on polymerizing MT plus-ends, and another that moves processively along MTs at high velocities—approximately seven times faster than the tip-tracking population. The tip-tracking pool may be regulated in a cell cycle-dependent manner, as dynein–dynactin tracking events appear less robust in mitotic cells compared to interphase cells. We propose that the ~3 µm run lengths measured for the motile pool likely underestimate the true processivity of dynein–dynactin. This is due to the relatively short and dynamic MT tracks available in interphase HeLa cells, and the fact that motile puncta could only be visualized while within the TIRF field and prior to photobleaching. Indeed, dynein was recently shown to be highly processive in iNeurons, with a mean run length of ~35 µm and some runs exceeding 100 µm (*Fellows et al., 2024*). In contrast, a separate study (*Tirumala et al., 2024*) reported that dynein is not highly processive, typically exhibiting runs of very short duration (~0.6 s) in HeLa cells. A notable technical difference that may account for this discrepancy is that our study visualizes endogenously tagged human DHC, whereas Tirumala et al. characterized over-expressed mouse DHC in HeLa cells. Overexpression of the DHC may result in an imbalance of the subunits that comprise the active motor complex, leading to inactive or less active complexes. Similarly, mouse DHC may not have the ability to efficiently assemble into active and processive dynein–dynactin–adaptor complexes to the same extent as human DHC. Two-color imaging of SiR-Tubulin-labeled MTs and dynein–dynactin revealed a population of p50-EGFP and DHC-EGFP puncta not evidently associated with MTs, suggesting the presence of a cortical pool of dynein–dynactin. Cortical dynein–dynactin has been clearly visualized in mitotic mammalian cells, where it functions in spindle positioning (*Busson et al., 1998*; *Collins et al., 2012*; *Faulkner et al., 2000*; *Kiyomitsu and Cheeseman, 2012*; *Kobayashi and Murayama, 2009*). In interphase mammalian cells, dynein–dynactin also contributes to centrosome positioning (*Etienne-Manneville and Hall, 2001*; *Palazzo et al., 2001*), and has been proposed to do so via pulling forces exerted on centrosomal MTs by a cortical pool (*Burakov et al., 2003*; *Dujardin and Vallee, 2002*; *Gundersen, 2002*) that we likely visualized by TIRFM. While a cortical pool of dynein–dynactin may exhibit lateral diffusion at the plasma membrane, it would, by necessity, be considerably less motile than running dynein–dynactin complexes.

The dynein–dynactin pool that exhibited directional motility on MTs was capable of switching MT tracks, which is a known behavior of dynein–dynactin (*Ross et al., 2008*), as evidenced by sharp changes in direction sometimes at near-perpendicular angles. Dynein–dynactin rarely (~3% of motile puncta) switched directions, suggesting that there is not a constant tug-of-war between dynein–dynactin and kinesins in proliferating cells as compared to observations of bidirectional vesicular transport in neurons (*Hancock, 2014*; *Hendricks et al., 2010*; *Maeder et al., 2014*). The low directional switching frequency observed here is consistent with in vitro studies reconstituting the tug-of-war phenomenon showing that one type of motor dominates once directional movement begins and that reversal events are rare (*Ali et al., 2023*; *Belyy et al., 2016*; *Derr et al., 2012*). The velocities measured here in HeLa cells were comparable to those measured for unopposed dynein and, therefore, suggest that there was not significant resistive drag from associated kinesin motors, as has been observed in vitro. The velocity and low switching frequency of motile puncta suggest that any kinesin motors associated with cargos being transported by the dynein–dynactin visualized here are inactive and/or cannot effectively bind the MT lattice during dynein–dynactin-mediated transport in interphase HeLa cells. Interestingly, directional switching was also uncommon during retrograde dynein–dynactin movements in iNeuron axons (*Fellows et al., 2024*) and in retrograde trafficking of early endosomes in *Ustilago maydis* mediated by dynein–dynactin containing the HOOK3 adaptor (*Bielska et al., 2014*; *Schuster et al., 2011*).

Finally, our cell-based data are consistent with recent in vitro characterizations of dynein–dynactin structure, function, and regulation (*Belyy et al., 2016*; *Chaaban and Carter, 2022*; *Urnavicius et al., 2018*). Based on our fluorescence intensity measurements, we favor the interpretation that the motile dynein–dynactin complexes visualized here are typically comprised of a single dynactin complex bound to a tandem array of two dyneins. Our measured dynein–dynactin velocity of 1.2 µm/s further supports

this conclusion since this speed is consistent with in vitro velocities of dynein–dynactin complexes containing activating adaptors that tend to recruit two dyneins. However, we do not exclude that the range of DHC velocities and intensities measured here may also include sub-populations of complexes containing a single dynein dimer. It would be valuable to test whether dynein–dynactin regulation mechanisms that have been characterized in vitro also apply to the physiological regulation of dynein–dynactin activity in living cells by measuring DHC motility parameters and intensities in cells depleted of different cargo adaptors. Ultimately, the direct visualization of human dynein–dynactin and quantification of its motility parameters and stoichiometries in living cells should be an important physiological complement to in vitro assays, which altogether will better inform mechanistic models of the cellular processes that rely on dynein–dynactin function.

# Materials and methods

## Key resources table

| Reagent type (species) or resource | Designation | Source or reference | Identifiers | Additional information |
|---|---|---|---|---|
| Gene [*Homo sapiens* (Human)] | Dynein cytoplasmic 1 heavy chain 1 (DYNC1H1) | HGNC:HGNC:2961 | Gene ID: 1778 | https://www.ncbi.nlm.nih.gov/gene/1778 |
| Gene [*Homo sapiens* (Human)] | Dynactin subunit 2 (DCTN2) | HGNC:HGNC:2712 | Gene ID: 10540 | https://www.ncbi.nlm.nih.gov/gene?Db=gene&Cmd=DetailsSearch&Term=10540 |
| Cell line (Human) | Parental HeLa | American Type Culture Collection (ATCC) | ATCC: CCL-2 | |
| Cell line (Human) | DHC (FKBP-EGFP-KI/+) KI = Knock-in | This study | | HeLa with tags added to the C-terminus of DYNC1H1/DHC |
| Cell line (Human) | p50(FKBP-EGFP-KI/+) KI = Knock-in | This study | | HeLa with tags added to the C-terminus of DCTN2/p50 |
| Cell line (*Drosophila melanogaster*) | *Drosophila* Schneider 2 (S2) cell line | American Type Culture Collection (ATCC) | ATCC: CRL-1963 | Also referred to as SL2 or DMel2 |
| Recombinant DNA reagent | pMT-Kinesin-1-EGFP | This study | | Modified from original construct in *Ye et al., 2018* |
| Antibody | Anti-dynactin p50 (mouse polyclonal) | BD Transduction Laboratories (Cat. # 611002) | | 1:1000 dilution used for western blots |
| Antibody | Donkey-anti-mouse IgG secondary antibodies conjugated with HRP | Jackson ImmunoResearch Laboratories, Inc (Code: **715-035-151**) | RRID:AB_2340771 | 1:5000 dilution used for western blots |
| Recombinant DNA reagent | pB80-hsKIF5B(1-560)-L-GFP | Addgene plasmid # 193716 | RRID:Addgene_193716 | |
| Recombinant DNA reagent | pSpCas9(BB)-2A-Puro (PX459) | Addgene plasmid # 62988 | RRID:Addgene_62988 | |
| Chemical compound, drug | SiR-Tubulin | Cytoskeleton | Cat. #CY-SC002 | |
| Chemical compound, drug | Antibiotic/antimycotic cocktail | Sigma-Aldrich | A5955-100ML | |
| Software, algorithm | MetaMorph | MetaMorph | | |
| Software, algorithm | ImajeJ,Fiji | NIH | | |
| Software, algorithm | MS office | Microsoft | | |
| Software, algorithm | Prism | GraphPad | | |
| Software, algorithm | R | https://www.R-project.org/ | | |
| Software, algorithm | PlotsOfDifferences | https://huygens.science.uva.nl/PlotsOfDifferences | | *Goedhart, 2019* |

## Cell culture

Parental HeLa and CRISPR-edited HeLa clones expressing DHC-EGFP or p50-EGFP were grown in standard DMEM medium (Gibco, USA) supplemented with 10% non-heat-inactivated fetal bovine serum (FBS; Gibco, USA) and 0.5× antibiotic/antimycotic cocktail (Sigma-Aldrich) and maintained at 37°C with 5% $CO_2$. Human-kinesin-1-EGFP-expressing *Drosophila* S2 cells were grown in Schneider's medium (Life Technologies) supplemented with 10% heat-inactivated FBS and 0.5× antibiotic/

antimycotic cocktail (Sigma), and maintained at 25°C. Kinesin-1-EGFP expression in S2 cells was induced with 500 µM $CuSO_4$ for 16–18 hr prior to imaging.

## CRISPR-engineered cell line production

### CRISPR gene editing

FKBP-EGFP tags were added to the C-terminus of human DHC and p50 using methods described previously (*Sheridan and Bentley, 2016*; *Stewart-Ornstein and Lahav, 2016*). In brief, repair cassettes comprised of FKBP-EGFP linked to a cleavable peptide (T2A) followed by a selectable marker (Neomycin) were cloned into pCMV and used for PCR reactions. Glycine–Alanine linkers were included between proteins in the construct. Guide sequences were selected using the CRISPR design tool (http://crispr.mit.edu/) from the Zhang lab at MIT (*Ran et al., 2013*) using 'other regions' and the human target genome (hg19). The search tool was used to select guides close to the C-terminus of the protein of interest (~100 nt surrounding and including the stop codon). Top and bottom oligos were obtained for each guide with the bases 5'-CACC-3' added to the top oligo and 5'-AAAC-3' added to the complement of the bottom oligo.

Guides were cloned into a Cas9-containing plasmid (PX459) Addgene #62988, (Cambridge, MA) following methods previously outlined (*Moyer and Holland, 2015*). Top and bottom oligos were annealed and then phosphorylated by T4 PNK (NEB, Ipswich, MA). Guides were then cloned into PX459 that was cut using BbsI (NEB, Ipswich, MA) and ligated using T4 ligase (NEB Ipswich, MA). Guide-Cas9-containing plasmids were then sequenced using the U6 promoter primer (*Ran et al., 2013*) and purified using either endotoxin-free mini-preps or midi-preps according to manufacturer protocol (Promega, Madison, WI). Repair cassettes were amplified using primers designed to be homologous to the C-terminal genomic DNA surrounding the STOP codon. In all cases, the guide target sequence was mutated to prevent Cas9 from recognizing the repair cassette.

Cells were grown in DMEM medium (Thermo Fisher Scientific) with 10% BS (Atlanta Biologicals, Flowery Branch, GA) and 0.5× antibiotic/antimycotic solution (final concentrations 50 U/ml penicillin, 0.05 mg/ml streptomycin, 0.125 µg/ml amphotericin B; Sigma-Aldrich, St. Louis, MO) at 37°C and 5% carbon dioxide ($CO_2$). For long-term storage, cells were frozen in DMEM medium with 5% FBS and 0.5× antibiotic/antimycotic solution and 15% DMSO and held at –80°C for 1–2 days before moving to liquid nitrogen.

To generate CRISPR-modified cell lines, parental cells were nucleofected using an Amaxa Nucleofector (Lonza, Portsmouth, NH) program I-013 and Mirus nucleofection reagent (Mirus Bio LLC, Madison, WI) according to the manufacturer's recommendations. Plasmids and Repair cassettes were used at a ratio of 1:1 at a concentration of 1 µg DNA each. Following nucleofection, cells were grown in regular growth media in 100 mm dishes for 48–72 hr. and then 0.2 g/L Neomycin/G418 (InvivoGen, San Diego, CA) selection media was added. Media was then changed daily for 10–14 days and colonies of green cells were picked using cloning rings and returned to regular media for further screening and experiments.

### Genotyping

Genomic DNA was isolated from clonal CRISPR-tagged cells using Genomic DNA Mini Kit (Invitrogen) according to the manufacturer's recommendations. DNA was then amplified by PCR using genomic primers targeting the Neomycin cassette and downstream of the Stop codon. Phusion polymerase (NEB) was used to amplify 1 µl of isolated genomic DNA in a 50-µl reaction for 30 cycles. The resulting PCR products were analyzed using 0.8% agarose gel electrophoresis. The larger band was excised, and gel extracted using QIAGEN gel extraction kit and sequenced to verify proper integration of the tag.

## Live-cell microscopy

Cells were seeded onto 35 mm glass bottom Petri dishes (Cellvis) 2–3 days prior to imaging. With the exception of the visualization of tip-tracking DHC-EGFP, cells were incubated in media supplemented with 1 µM SiR-Tubulin (Cytoskeleton, Inc) for 30–60 min prior to washing out the SiR-Tubulin with fresh DMEM before imaging. The cells were visualized on a Nikon Ti-E microscope equipped with a TIRF illuminator, Borealis (Andor) retrofitted CSU-10 (Yokogawa) spinning disc head, a 100× 1.49-NA differential interference contrast TIRF Apochromat oil-immersion objective, Nikon perfect

focus system (PFS), two ORCA-Flash4.0 LT Digital CMOS cameras (Hamamatsu), four laser lines (447, 488, 561, and 641 nm) in a dual-output laser launch system (Andor), and MetaMorph software (Molecular Devices). The imaging mode (spinning disc versus TIRFM) of the system is set by selecting the appropriate fiber optic output and corresponding CMOS camera. For TIRFM imaging, interphase cells were identified and the initial focal plane was set by focusing on the far-red SiR tubulin signal with a particular emphasis on finding cells in which individual MTs could be seen between the nucleus and the glass-adhered plasma membrane. Individual snapshots were then taken on the EGFP channel to refine the focal plane on the DHC or p50 puncta. Once individual puncta were well-resolved, the PFS was engaged and the cell was subjected to streaming TIRFM for 12 s at an acquisition rate of 5 frames per second (200-ms exposure) and 2 × 2 camera binning. The population of dynein–dynactin on MTs positioned between the plasma membrane and nucleus would fall within the 100–200 nm excitable range of the evanescent wave produced by TIRF since the plasma membrane is typically ~5 nm and an MT is ~25 nm. While TIRFM is capable of effectively visualizing dynein–dynactin motility in living cells, in some time-lapses the excitation laser may have been angled to propagate as a highly inclined and laminated optical sheet (HILO) (*Tokunaga et al., 2008*). The experimental design and analyses would be unaffected by whether the motile punta were visualized via TIRFM or HILO microscopy. For consistency, the imaging mode employed in this study is referred to as TIRFM.

## Quantifications of motility parameters and fluorescence intensities

High temporal (5 fps) TIRFM time-lapses were examined by eye to identify evident motile puncta, the paths of which were traced manually in MetaMorph using the Multi-Line drawing function. A kymograph of the line segment with a 5-pixel line width was then generated and the velocity was determined by measuring the slope of the motility event on the kymograph from start to end using the Single Line drawing function. The run time was measured by defining the start and end frame of evident movement in the TIRFM time-lapse, and the run length was then calculated by multiplying the run time by the velocity measured in the kymograph of the motility event. The run times and run lengths are likely underestimates of what dynein–dynactin can achieve due to the fact that the puncta sometimes bleached or exited the TIRFM evanescent field.

The number of EGFP molecules per motile puncta of DHC and p50 was determined by comparing their background-corrected fluorescence intensities to that of a known dimer standard: human kinesin-1 tagged with EGFP in both *Drosophila* S2 cells and in HeLa cells. As previously described (*Ye et al., 2018*), *Drosophila* S2 cells expressing inducible human kinesin-1 (in this case tagged with EGFP) were induced with 500 µM CuSO4 for 16 hr to induce expression. The next day, the induced S2 cells were seeded onto a concanavalin A coated glass bottom Petri dish and allowed to adhere for ~1 hr prior to visualization by TIRFM. The DHC-EGFP and p50-EGFP HeLa clones and the human kinesin-1-EGFP-expressing S2 cells were each visualized sequentially via live-cell TIRF microscopy using identical imaging parameters (5 fps acquisition rate, 200-ms exposure, 2 × 2 binning) within the same region of the camera chip.

To compare DHC-EGFP and p50-EGFP intensities to human kinesin-1-EGFP in HeLa cells, HeLa cells were seeded in a 35-mm dish in 2 ml of complete HeLa medium (DMEM supplemented with 10% FBS and 0.5× antibiotic–antimycotic mix) 18–24 hr prior to transfection at an initial density ($0.25 \times 10^6$) to achieve ~50% confluency on the day of transfection. On the day of transfection, two sterile 1.5 ml microcentrifuge tubes were prepared and labeled A and B. Each tube received 125 µl of OPTI-MEM medium. Tube A was supplemented with 7.5 µl of Lipofectamine 3000 reagent and vortexed briefly. Tube B was supplemented with 250 ng of endotoxin-free plasmid pB80-hsKIF5B(1-560)-L-GFP (human kinesin-1-EGFP) (Addgene plasmid # 193716) and 4 µl of P3000 reagent, followed by brief vortexing. The contents of tube B were then transferred to tube A, vortexed gently, and incubated at room temperature for 15 min to allow complex formation. The resulting DNA–lipid complexes were added dropwise to the HeLa cells, which were then incubated at 37°C in a humidified $CO_2$ incubator for 24 hr. After incubation, the medium was replaced with 2 ml of fresh HeLa medium, and cells were subjected to imaging over the next 1–2 days (2–3 days post-transfection). The kinesin-1-EGFP transfected cells, and DHC-EGFP and p50-EGFP HeLa clones were each visualized sequentially on the same day via live-cell TIRF microscopy using identical imaging parameters within the same region of the camera chip. The region-in-region method (*Ye and Maresca, 2018*) was used to background correct and quantify the integrated fluorescence intensities of motile puncta of DHC-, p50-, and kinesin-1-EGFP (for both

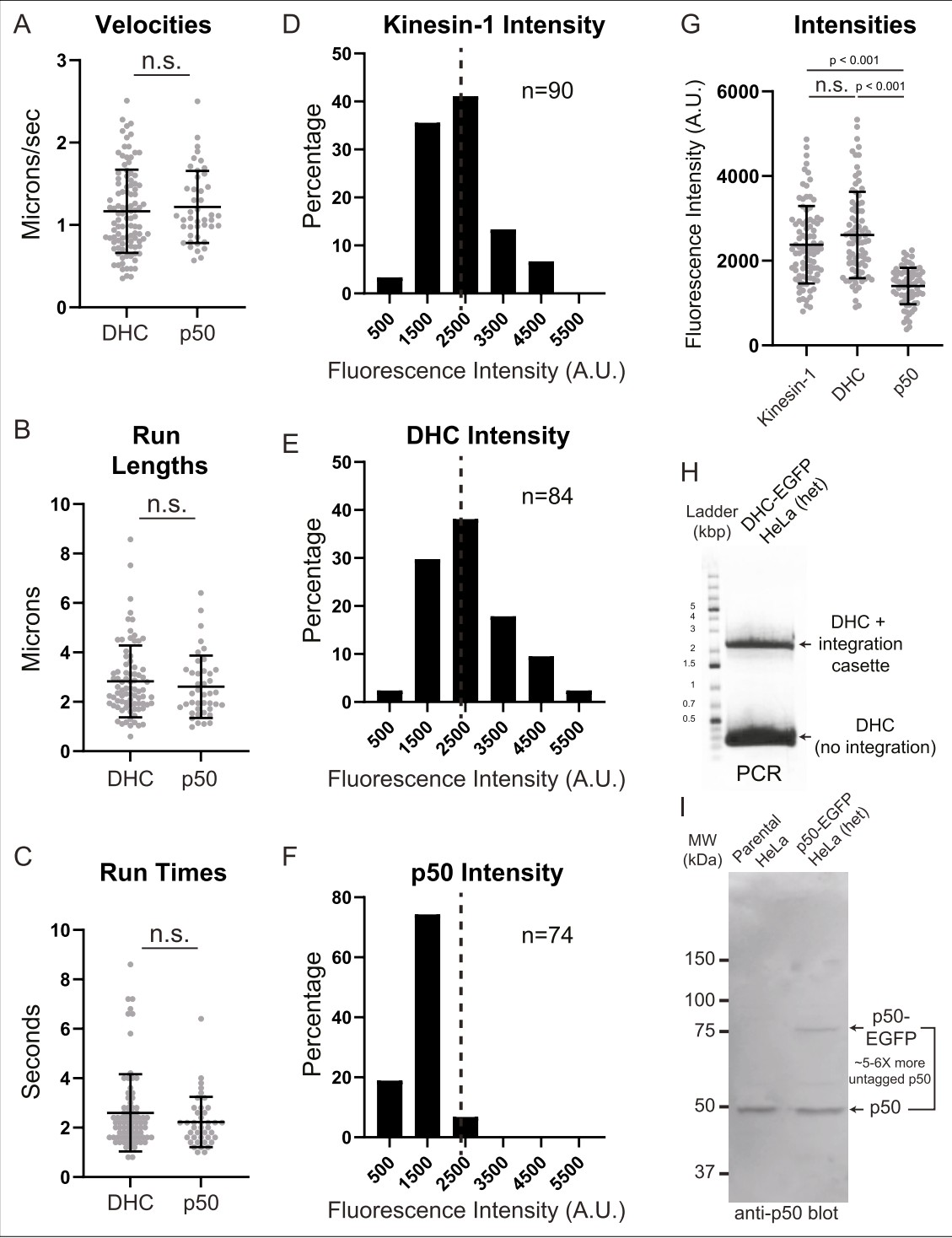

**Figure 3.** Motile DHC and p50 puncta exhibit identical motility parameters but different fluorescent intensities. Scatter plots of (**A**) velocities (DHC, *n* = 100; p50, *n* = 44), (**B**) run lengths (DHC, *n* = 81; p50, *n* = 41), and (**C**) run times (DHC, *n* = 81; p50, *n* = 41). Distributions of the background-corrected fluorescence intensities of motile puncta of (**D**) kinesin-1-EGFP transiently expressed in HeLa cells, (**E**) DHC-EGFP, and (**F**) p50-EGFP. The dashed line in each histogram denotes the mean value of the kinesin-1-EGFP dataset. (**G**) Scatter plots of the kinesin-1, DHC, and p50 fluorescence intensities (kinesin-1, *n* = 90 puncta; DHC, *n* = 84 puncta; p50, *n* = 74 puncta). (**H**) PCR of genomic DNA from the DHC-EGP clone used in this study using PCR primers flanking the integration site of the repair cassette. The upper band was extracted and subjected to sequencing, the results of which are shown in *Figure 1—figure supplement 1A*. (**I**) Western blot for p50 of cell lysates from the parental HeLa cell line and the p50-EGFP clone used in this study. The tagged p50 runs ~30 kDa larger than the untagged p50 and is expressed at ~5- to 6-fold lower levels than the endogenous p50. Error bars are mean values ± standard deviations. The reported p-values were determined by a randomization method: n.s. is not significant (p > 0.05).

*Figure 3 continued on next page*

*Figure 3 continued*

The online version of this article includes the following source data and figure supplement(s) for figure 3:

**Source data 1.** PowerPoint file containing original image of agarose gel for *Figure 3H*, indicating the relevant PCR fragments.

**Source data 2.** Original file of agarose gel image in *Figure 3H*.

**Source data 3.** PowerPoint file containing original membrane and western blots for *Figure 3I*, indicating the relevant bands and cell line lysates.

**Source data 4.** Original files for western blot in *Figure 3I*.

**Source data 5.** Excel spreadsheet containing the underlying processed data and numerical values for plots in *Figure 3*.

**Figure supplement 1.** Motility parameters of the transiently expressed Kinesin-1-EGFP in HeLa cells.

**Figure supplement 1—source data 1.** Excel spreadsheet containing the underlying data and numerical values for plots in *Figure 3—figure supplement 1*.

**Figure supplement 2.** Fluorescence intensity comparisons to Kinesin-1-EGFP transiently expressed in *Drosophila* S2 cells.

**Figure supplement 2—source data 1.** Excel spreadsheet containing the underlying data and numerical values for plots in *Figure 3—figure supplement 2*.

---

the S2 cell and HeLa cell comparisons). The intensity of each motile puncta was measured for a single time point from each run in which the spot was most clearly resolved and did not have significant local background signal from nearby EGFP puncta. Cytoplasmic levels of DHC-EGFP and p50-EGFP were quantified from max projections of spinning disk confocal Z-stacks by subtracting the integrated intensity of a 25 × 25 pixel square ROI in the nucleus from the integrated intensity of a 25 × 25 pixel ROI placed in a representative region of the cytoplasm.

## Western blotting

Twenty µg of total protein from cell lysates prepared from parental HeLa cells or the p50-EGFP clone were loaded onto a 10% SDS–PAGE gel, run out, and transferred to a nitrocellulose membrane on the Trans-Blot Turbo transfer system (Bio-Rad Laboratories) using the preprogrammed 'HIGH MW' 10-min protocol. After blocking for 1 hr in TBS with 0.1% Tween and 5% milk, the membrane was incubated overnight at 4°C with mouse anti-dynactin p50 (BD Transduction Laboratories, Cat. # 611002) diluted at 1:1000 in the block. The membrane was washed 3 × 5 min in TBS + 0.1% Tween and then incubated for 1 hr in milk containing donkey-anti-mouse IgG secondary antibodies conjugated with HRP (Jackson ImmunoResearch Laboratories, Inc) diluted at 1:5000 in block. Following 3 × 5 min washes in TBS + 0.1% Tween, the blot was incubated with Immobilon Western Chemiluminescent HRP Substrate (Millipore) according to the manufacturer's recommendations. Imaging of the membrane was done on a G:Box system controlled by GeneSnap software (Syngene).

## Cell lines

The parental HeLa cells, sourced from American Type Culture Collection (ATCC), and both CRISPR knock-in HeLa cell lines (DHC-EGFP and p50-EGFP) that were generated from the parental HeLa cell line were authenticated via STR profiling (ATCC). Production of EGFP-tagged p50 in the knock-in cell line was confirmed via western blot for p50 and integration of the EGFP repair cassette in the DHC-EGFP knock-in cell line was confirmed by amplicon sequencing (Plasmidsaurus). Stocks of the parental and knock-in cell lines tested negative for mycoplasma (ATCC) shortly before publication.

## Statistical analyses

Reported p-values were generated with a randomization method using the PlotsOfDifferences web tool at https://huygens.science.uva.nl/PlotsOfDifferences (*Goedhart, 2019*). PlotsOfDifferences does not rely on assumptions about the distribution of the data (normal versus non-normal) to calculate p-values.

## Acknowledgements

We are grateful to Barbara Mann for generating the CRISPR-engineered HeLa cell lines. Thank you to Thomas Laskarzewski for help with R and data visualization. Thank you to Carline Fermino do Rosário for assistance with cell line authentication and mycoplasma testing. pB80-hsKIF5B(1-560)-L-GFP was a gift from Lukas Kapitein (Addgene plasmid # 193716; http://n2t.net/addgene:193716;

RRID:Addgene_193716). This work was supported by NIH grants (GM107026 and GM156188) to TJM and by an NSF grant (MCB1817926) to PW.

## Additional information

### Funding

| Funder | Grant reference number | Author |
|---|---|---|
| National Institutes of Health | GM107026 | Thomas J Maresca |
| National Institutes of Health | GM156188 | Thomas J Maresca |
| National Science Foundation | MCB1817926 | Patricia Wadsworth |

The funders had no role in study design, data collection, and interpretation, or the decision to submit the work for publication.

### Author contributions

Vikash Verma, Formal analysis, Investigation, Visualization, Methodology, Writing – review and editing; Patricia Wadsworth, Conceptualization, Resources, Supervision, Funding acquisition, Methodology, Writing – original draft, Project administration, Writing – review and editing; Thomas J Maresca, Conceptualization, Resources, Data curation, Formal analysis, Supervision, Funding acquisition, Investigation, Visualization, Methodology, Writing – original draft, Project administration, Writing – review and editing

### Author ORCIDs

Vikash Verma https://orcid.org/0000-0003-2371-4164
Patricia Wadsworth https://orcid.org/0000-0003-1364-7893
Thomas J Maresca https://orcid.org/0000-0003-2214-8674

Reviewer #1 (Public Review): https://doi.org/10.7554/eLife.94963.3.sa1
Reviewer #2 (Public Review): https://doi.org/10.7554/eLife.94963.3.sa2
Reviewer #3 (Public Review): https://doi.org/10.7554/eLife.94963.3.sa3
Author response https://doi.org/10.7554/eLife.94963.3.sa4

## Additional files

### Supplementary files

MDAR checklist

### Data availability

Source data used for generating the figures has been provided.

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
