## [Editor Report · eLife Assessment]

In this **valuable** technical report, Verma et al. provide **convincing** evidence that endogenously tagged dynein and dynactin form processive motor complexes that move along microtubules in living cells. Using quantitative fluorescence microscopy, they directly compare the stoichiometry and motility of these complexes to kinesin-1, revealing distinct transport behaviors and regulatory properties. This study offers key methodological and conceptual advance for understanding the dynamics of native motor proteins within the cellular environment and will be of interest to the cell biology community.

---

## [Referee Report · Reviewer #1 (Public Review)]

The manuscript by Verma et al. is a simple and concise assessment of the in-cell motility parameters of cytoplasmic dynein. Although numerous studies have focused on understanding the mechanism by which dynein is activated using a complement of in vitro methodologies, an assessment of dynein motility in cells has been lacking. It has been unclear whether dynein exhibits high processivity within the crowded and complicated environment of the cell. For example, does cargo-bound dynein exhibit short, non-processive motility (as has been recently suggested; Tirumala et al., 2022 bioRxiv)? Does cargo-bound dynein move against opposing forces generated by cargo-bound kinesins? Do cargoes exhibit bidirectional switching due to stochastic activation of kinesins and dyneins? The current work addresses these questions quite simply by observing and quantitating the motility of natively tagged dynein in HeLa cells.

---

## [Referee Report · Reviewer #2 (Public Review)]

Verma et al. provide a short technical report showing that endogenously tagged dynein and dynactin molecules localize to growing microtubule plus-ends and also move processively along microtubules in cells. The data are convincing, and the imaging and movies very nicely demonstrate their claims. I don't have any large technical concerns about the work. It is perhaps not surprising that dynein-dynactin complexes behave this way in cells due to other reports on the topic, but the current data are among some of the nicest direct demonstrations of this phenomenon. It may be somewhat controversial since a separate group has reported that dynein does not move processively in mammalian cells

(https://www.biorxiv.org/content/10.1101/2021.04.05.438428v3).

---

## [Referee Report · Reviewer #3 (Public Review)]

In this manuscript, Verma et al. set out to visualize cytoplasmic dynein in living cells and describe their behaviour. They first generated heterozygous CRISPR-Cas9 knock-ins of DHC1 and p50 subunit of dynactin and used spinning disk confocal microscopy and TIRF microscopy to visualize these EGFP-tagged molecules. They describe robust localization and movement of DHC and p50 at the plus tips of MTs, which was abrogated using SiR tubulin to visualize the pool of DHC and p50 on the MTs. These DHC and p50 punctae on the MTs showed similar, highly processive movement on MTs. Based on comparison to inducible EGFP-tagged kinesin-1 intensity in Drosophila S2 cells, the authors concluded that the DHC and p50 punctae visualized represented 1 DHC-EGFP dimer+1 untagged DHC dimer and 1 p50-EGFP+3 untagged p50 molecules.

---

## [Author Response]

The following is the authors’ response to the original reviews.

**Reviewer #1 (Public Review):**
Strengths:The work uses a simple and straightforward approach to address the question at hand: is dynein a processive motor in cells? Using a combination of TIRF and spinning disc confocal microscopy, the authors provide a clear and unambiguous answer to this question.

Thank you for the recognition of the strength of our work

Weaknesses:My only significant concern (which is quite minor) is that the authors focus their analysis on dynein movement in cells treated with docetaxol, which could potentially affect the observed behavior. However, this is likely necessary, as without it, motility would not have been observed due to the 'messiness' of dynein localization in a typical cell (e.g., plus end-tracking in addition to cargo transport).

You are exactly correct that this treatment was required to provided us a clear view of motile dynein and p50 puncta. One concern about the treatment that we had noted in our original submission was that the docetaxel derivative SiR tubulin could increase microtubule detyrosination, which has been implicated in affecting the initiation of dynein-dynactin motility but not motility rates (doi: 10.15252/embj.201593071). In response to a comment from reviewer 2 we investigated whether there was a significant increase in alpha-tubulin detyrosination in our treatment conditions and found that there was not. We have removed the discussion of this possibility from the revised version. Please also see response to comments raised by reviewer 2.

**Reviewer 1 (Recommendations for the authors):**
Major points:(1) The authors measured kinesin-1-GFP intensities in a different cell line (drosophila S2 cells) than what was used for the DHC and p50 measurements (HeLa cells). It is unclear if this provides a fair comparison given the cells provide different environments for the GFP. Although the differences may in fact be trivial, without somehow showing this is indeed a fair comparison, it should at least be noted as a caveat when interpreting relative intensity differences. Alternatively, the authors could compare DHC and p50 intensities to those measured from HeLa cells treated with taxol.

Thank you for this suggestion. We conducted new rounds of imaging with the DHCEGFP and p50-EGFP clones in conjunction with HeLa cells transiently expressing the human kinesin-1-EGFP and now present the datasets from the new experiments. Importantly, our new data was entirely consistent with the prior analyses as there was not a significant difference between the kinesin-1-EGFP dimer intensities and the DHC-EGFP puncta intensities and there was a statistically significant difference in the intensity of p50 puncta, which were approximately half the intensity of the kinesin-1 and DHC. We have moved the old data comparing the intensities in S2 cells expressing kinesin-1-EGFP to Figure 3 - figure supplement 2 A-D and the new HeLa cell data is now shown in Figure 3 D-G.

(2) Given the low number of observations (41-100 puncta), I think a scatter plot showing all data points would offer readers a more transparent means of viewing the single-molecule data presented in Figures 3A, B, C, and G. I also didn't see 'n' values for plots shown in Figure 3.

The box and whisker plots have now been replaced with scatter plots showing all data points. The accompanying ‘n’ values have been included in the figure 3 legend as well as the histograms in figures 1 and 2 that are represented in the comparative scatter plots.

(3) Given the authors have produced a body of work that challenges conclusions from another pre-print (Tirumala et al., 2022 bioRxiv) - specifically, that dynein is not processive in cells - I think it would be useful to include a short discussion about how their work challenges theirs. For example, one significant difference between the two experimental systems that may account for the different observations could simply be that the authors of the Tirumala study used a mouse DHC (in HeLa cells), which may not have the ability to assemble into active and processive dynein-dynactin-adaptor complexes.

Thank you for pointing this out! At the time we submitted our manuscript we were conflicted about citing a pre-print that had not been peer reviewed simply to point out the discrepancy. If we had done so at that time we would have proposed the exact potential technical issue that you have proposed here. However, at the time we felt it would be better for these issues to be addressed through the review process. Needless to say, we agree with your interpretation and now that the work is published (Tirumala et al. JCB, 2024) it is entirely appropriate to add a discussion on Tirumala et al. where contradictory observations were reported.

The following statement has been added to the manuscript:

“In contrast, a separate study (Tirumala et al., 2024) reported that dynein is not highly processive, typically exhibiting runs of very short duration (~0.6 s) in HeLa cells. A notable technical difference that may account for this discrepancy is that our study visualizes endogenously tagged human DHC, whereas Tirumala et al. characterized over-expressed mouse DHC in HeLa cells. Over-expression of the DHC may result in an imbalance of the subunits that comprise the active motor complex, leading to inactive, or less active complexes. Similarly, mouse DHC may not have the ability to efficiently assemble into active and processive dynein-dynactin-adaptor complexes to the same extent as human DHC.”

Minor points:(1) "Specifically, the adaptor BICD2 recruited a single dynein to dynactin while BICDR1 and HOOK3 supported assembly of a "double dynein" complex." It would be more accurate to say that dynein-dynactin complexes assembled with Bicd2 "tend to favor single dynein, and the Bicdr1 and Hook3 tend to favor two dyneins" since even Bicd2 can support assembly of 2 dynein-1 dynactin complexes (see Urnavicius et al, Nature 2018).

Thank you, the manuscript has been edited to reflect this point.

(2) "Human HeLa cells were engineered using CRISPR/Cas9 to insert a cassette encoding FKBP and EGFP tags in the frame at the 3' end of the dynein heavy chain (DYNC1H1) gene (SF1)." It is unclear to what "SF1" is referring.

SF1 is supplementary figure 1, which we have now clarified as being Figure 1 – figure supplement 1A.

(3) "The SiR-Tubulin-treated cells were subjected to two-color TIRFM to determine if the DHC puncta exhibited motility and; indeed, puncta were observed streaming along MTs..." This sentence is strangely punctuated (the ";" is likely a typo?).

Thank you for pointing this out, the typo has been corrected and the sentence now reads:

“The SiR-Tubulin-treated cells were subjected to two-color TIRFM and DHC-EGFP puncta were clearly observed streaming on Sir-Tubulin labeled MTs, which was especially evident on MTs that were pinned between the nucleus and the plasma membrane (Video 3)”

(4) I am unfamiliar with the "MK" acronym shown above the molecular weight ladders in Figure 3H and I. Did the authors mean to use "MW" for molecular weight?

We intended this to mean MW and the typo has been corrected.

(5) "This suggests that the cargos, which we presume motile dynein-dynactin puncta are bound to, any kinesins..." This sentence is confusing as written. Did the authors mean "and kinesins"?

Agreed. We have changed this sentence to now read:

“The velocity and low switching frequency of motile puncta suggest that any kinesin motors associated with cargos being transported by the dynein-dynactin visualized here are inactive and/or cannot effectively bind the MT lattice during dynein-dynactin-mediated transport in interphase HeLa cells.”

**Reviewer 2 (Recommendations for the authors):**
(1) I am confused as to why the authors introduced an FKBP tag to the DHC and no explanation is given. Is it possible this tag induces artificial dimerization of the DHC?

FKBP was tagged to DHC for potential knock sideways experiments. Since the current cell line does not express the FKBP counterpart FRB, having FKBP alone in the cell line would not lead to artificial dimerization of DHC.

(2) The authors use a high concentration of SiR-tubulin (1uM) before washing it out. However, they observe strong effects on MT dynamics. The manufacturer states that concentrations below 100nM don't affect MT dynamics, so I am wondering why the authors are using such a high amount that leads to cellular phenotypes.

We would like to note that in our hands even 100 nM SiR-tubulin impacted MT dynamics if it was incubated for enough time to get a bright signal for imaging, which makes sense since drugs like docetaxel and taxol become enriched in cells over time. Thus, it was a trade-off between the extent/brightness of labeling and the effects on MT dynamics. We opted for shorter incubation with a higher concentration of Sir-Tubulin to achieve rapid MT labeling and efficient suppression of plus-end MT polymerization. This approach proved useful for our needs since the loss of the tip-tacking pool of DHC provided a clearer view of the motile population of MT-associated DHC.

(3) The individual channels should be labeled in the supplemental movies.

They have now been labelled.

(4) I would like to see example images and kymographs of the GFP-Kinesin-1 control used for fluorescent intensity analysis. Further, the authors use the mean of the intensity distribution, but I wonder why they don't fit the distribution to a Gaussian instead, as that seems more common in the field to me. Do the data fit well to a Gaussian distribution?

Example images and kymographs of the kinesin-1-EGFP control HeLa cells used for the updated fluorescent intensity analysis have been now added to the manuscript in Figure 3 - figure supplement 1. The kinesin-1-EGFP transiently expressed in HeLa cells exhibited a slower mean velocity and run length than the endogenously tagged HeLa dynein-dynactin. Regarding the distribution, we applied 6 normality tests to the new datasets acquired with DHC and p50 in comparison to human kinesin-EGFP in HeLa cells. While we are confident concluding that the data for p50 was normally distributed (p > 0.05 in 6/6), it was more difficult to reach conclusions about the normality of the datasets for kinesin-1 (p > 0.05 in 4/6) and DHC (p > 0.5 in 1/6). We have decided to report the data as scatter plots (per the suggestion in major point 1 by reviewer 1) in the new Figure 3G since it could be misleading to fit a non-normal distribution with a single Gaussian. We note that the likely non-normal distribution of the DHC data (since it “passed” only 1/6 normality tests) could reflect the presence of other populations (e.g. 1 DHC-EGFP in a motile puncta), but we could also not confidently conclude this since attempting to fit the data with a double Gaussian did not pass statistical muster. Indeed, as stated in the text, on lines 197-198 we do not exclude that the range of DHC intensities measured here may include sub-populations of complexes containing a single dynein dimer with one DHC-EGFP molecule.

Ultimately, we feel the safest conclusion is that there was not a statically significant difference between the DHC and kinesin-1 dimers (p = 0.32) but there was a statistically significant difference between both the DHC and kinesin-1 dimers compared to the p50 (p values < 0.001), which was ~50% the intensity of DHC and kinesin-1. Altogether this leads us to the fairly conservative conclusion that DHC puncta contain at least one dimer while the p50 puncta likely contain a single p50-EGFP molecule.

(5) The authors suggest the microtubules in the cells treated with SiR-tubulin may be more detyrosinated due to the treatment. Why don't they measure this using well-characterized antibodies that distinguish tyrosinated/detyrosinated microtubules in cells treated or not with SiR-tubulin?

At your suggestion, we carried out the experiment and found that under our labeling conditions there was not a notable difference in microtubule detyrosination between DMSO- and SiR-Tubulin-treated cells. Thus, we have removed this caveat from the revised manuscript.

(6) "While we were unable to assess the relative expression levels of tagged versus untagged DHC for technical reasons." Please describe the technical reasons for the inability to measure DHC expression levels for the reader.

We made several attempts to quantify the relative amounts of untagged and tagged protein by Western blotting. The high molecular weight of DHC (~500kDa) makes it difficult to resolve it on a conventional mini gel. We attempted running a gradient mini gel (4%-15%), and doing a western blot; however, we were still unable to detect DHC. To troubleshoot, the experiments were repeated with different dilutions of a commercially available antibody and varying concentrations of cell lysate; however, we were unable to obtain a satisfactory result.

We hold the view that even if it had it worked it would have been difficult to detect a relatively small difference between the untagged (MW = 500kDa) and tagged DHC (MW = 527kDa) by western blot. We have added language to this effect in the revised manuscript.

**Reviewer #3 (Public Review):**
(1). CRISPR-edited HeLa clones:(i) The authors indicate that both the DHC-EGFP and p50-EGFP lines are heterozygous and that the level of DHC-EGFP was not measured due to technical difficulties. However, quantification of the relative amounts of untagged and tagged DHC needs to be performed - either using Western blot, immunofluorescence or qPCR comparing the parent cell line and the cell lines used in this work.

See response to reviewer 2 above.

(ii) The localization of DHC predominantly at the plus tips (Fig. 1A) is at odds with other work where endogenous or close-to-endogenous levels of DHC were visualized in HeLa cells and other non-polarized cells like HEK293, A-431 and U-251MG (e.g.: OpenCell (https://opencell.czbiohub.org/target/CID001880), Human Protein Atlas), https://www.biorxiv.org/content/10.1101/2021.04.05.438428v3. The authors should perform immunofluorescence of DHC in the parental cells and DHC-EGFP cells to confirm there are no expression artifacts in the latter. Additionally, a comparison of the colocalization of DHC with EB1 in the parental and DHC-EGFP and p50-EGFP lines would be good to confirm MT plus-tip localisation of DHC in both lines.

The microtubule (MT) plus-tip localization of DHC was already observed in the 1990s, as evidenced by publications such as (PMID:10212138) and (PMID:12119357), which were further confirmed by Kobayashi and Murayama in 2009 (PMID:19915671). We hold the view that further investigation into this localization is not worthwhile since the tip-tracking behavior of DHC-dynactin has been long-established in the field.

(iii) It would also be useful to see entire fields of view of cells expressing DHC-EGFP and p50EGFP (e.g. in Spinning Disk microscopy) to understand if there is heterogeneity in expression. Similarly, it would be useful to report the relative levels of expression of EGFP (by measuring the total intensity of EGFP fluorescence per cell) in those cells employed for the analysis in the manuscript.

Representative images of fields have been added as Figure 1 - figure supplement 1B and Figure 2 – figure supplement 1 in the revised manuscript. We did not see drastic cell-tocell variation of expression within the clonal cell lines.

(iv) Given that the authors suspect there is differential gene regulation in their CRISPR-edited lines, it cannot be concluded that the DHC-EGFP and p50-EGFP punctae tracked are functional and not piggybacking on untagged proteins. The authors could use the FKBP part of the FKBPEGFP tag to perform knock-sideways of the DHC and p50 to the plasma membrane and confirm abrogation of dynein activity by visualizing known dynein targets such as the Golgi (Golgi should disperse following recruitment of EGFP-tagged DHC-EGFP or p50-EGFP to the PM), or EGF (movement towards the cell center should cease).

Despite trying different concentrations and extensive troubleshooting, we were not able to replicate the reported observations of Ciliobrevin D or Dynarrestin during mitosis. We would like to emphasize that the velocity (1.2 μm/s) of dynein-dynactin complexes that we measured in HeLa cells was comparable to those measured in iNeurons by Fellows et al. (PMID: 38407313) and for unopposed dynein under in vitro conditions.

(2) TIFRM and analysis:(i) What was the rationale for using TIRFM given its limitation of visualization at/near the plasma membrane? Are the authors confident they are in TIRF mode and not HILO, which would fit with the representative images shown in the manuscript?

To avoid overcrowding, it was important to image the MT tracks that that were pinned between the nucleus and the plasma membrane**.** It is unclear to us why the reviewer feels that true TIRFM could not be used to visualize the movement of dynein-dynactin on this population of MTs since the plasma membrane is ~ 3-5 nm and a MT is ~25-27 nm all of which would fall well within the 100-200 nm excitable range of the evanescent wave produced by TIRF. While we feel TIRF can effectively visualize dynein-dynactin motility in cells, we have mentioned the possibility that some imaging may be HILO microscopy in the materials and methods.

(ii) At what depth are the authors imaging DHC-EGFP and p50-EGFP?

The imaging depth of traditional TIRFM is limited to around 100-200 nm. In adherent interphase HeLa cells the nucleus is in very close proximity (nanometer not micron scale) to the plasma membrane with some cytoskeletal filaments (actin) and microtubules positioned between the plasma membrane and the nuclear membrane. The fact that we were often visualizing MTs positioned between the nucleus and the membrane makes us confident that we were imaging at a depth (100 - 200nm) consistent with TIRFM.

(iii) The authors rely on manual inspection of tracks before analyzing them in kymographs - this is not rigorous and is prone to bias. They should instead track the molecules using single particle tracking tools (eg. TrackMate/uTrack), and use these traces to then quantify the displacement, velocity, and run-time.

Although automated single particle tracking tools offer several benefits, including reduced human effort, and scalability for large datasets, they often rely on specialized training datasets and do not generalize well to every dataset. The authors contend that under complex cellular environments human intervention is often necessary to achieve a reliable dataset. Considering the nature of our data we felt it was necessary to manually process the time-lapses.

(iv) It is unclear how the tracks that were eventually used in the quantification were chosen. Are they representative of the kind of movements seen? Kymographs of dynein movement along an entire MT/cell needs to be shown and all punctae that appear on MTs need to be tracked, and their movement quantified.

Considering the densely populated environment of a cell, it will be nearly impossible to quantity all the datasets. We selected tracks for quantification, focusing on areas where MTs were pinned between the nucleus and plasma membrane where we could track the movement of a single dynein molecule and where the surroundings were relatively less crowded.

(v) What is the directionality of the moving punctae?

In our experience, cells rarely organized their MTs in the textbook radial MT array meaning that one could not confidently conclude that “inward” movements were minus-end directed. Microtubule polarity was also not able to be determined for the MTs positioned between the plasma membrane and the nucleus on which many of the puncta we quantified were moving. It was clear that motile puncta moving on the same MT moved in the same direction with the exception of rare and brief directional switching events. What was more common than directional switching on the same MT were motile puncta exhibiting changes in direction at sharp (sometimes perpendicular) angles indicative of MT track switching, which is a well-characterized behavior of dynein-dynactin (See DOI: 10.1529/biophysj.107.120014).

(vi) Since all the quantification was performed on SiR tubulin-treated cells, it is unclear if the behavior of dynein observed here reflects the behavior of dynein in untreated cells. Analysis of untreated cells is required.

It was important to quantify SiR tubulin-treated cells because SiR-Tubulin is a docetaxel derivative, and its addition suppressed plus-end MT polymerization resulting in a significant reduction in the DHC tip-tracking population and a clearer view of the motile population of MT-associated DHC puncta. Otherwise, it was challenging to reliably identify motile puncta given the abundance of DHC tip-tracking populations in untreated cells.

(3) Estimation of stoichiometry of DHC and p50Given that the punctae of DHC-EGFP and p50 seemingly bleach on MT before the end of the movie, the authors should use photobleaching to estimate the number of molecules in their punctae, either by simple counting the number of bleaching steps or by measuring single-step sizes and estimating the number of molecules from the intensity of punctae in the first frame.

Comparing the fluorescence intensity of a known molecule (in our case a kinesin-1EGFP dimer) to calculate the numbers of an unknown protein molecule (in our case Dynein or p50) is a widely accepted technique in the field. For example, refer to PMID: 29899040. To accurately estimate the stoichiometry of DHC and p50 and address the concerns raised by other reviewers, we expressed the human kinesin-EGFP in HeLa cells and analyzed the datasets from new experiments. We did not observe any significant differences between our old and new datasets.

(4) Discussion of prior literatureRecent work visualizing the behavior of dyneins in HeLa cells (DOI: 10.1101/2021.04.05.438428), which shows results that do not align with observations in this manuscript, has not been discussed. These contradictory findings need to be discussed, and a more objective assessment of the literature in general needs to be undertaken.